# Technical note: A stochastic framework for identification and evaluation of flash drought

Yuxin Li[1,2], Sisi Chen[1,2], Jun Yin[1,2], and Xing Yuan[1,2]

[1]Key Laboratory of Hydrometeorological Disaster Mechanism and Warning of Ministry of Water Resources/Collaborative Innovation Center on Forecast and Evaluation of Meteorological Disasters, Nanjing University of Information Science and Technology, Nanjing, 210044, China
[2]School of Hydrology and Water Resources, Nanjing University of Information Science and Technology, Nanjing, China

**Correspondence:** J.Y. (jun.yin@nuist.edu.cn)

**Abstract.** The rapid development of droughts, referred to as flash droughts, can pose serious impacts on agriculture, ecosystem, human health, and society. However, its definition using pentad-average soil moisture could result in low accuracy in assessing the drought occurrence, making it difficult to analyze various factors controlling the formation of flash drought. Here we used a stochastic water balance framework to quantify the whole probability structure of the timing for soil moisture dropping from a higher level to a lower one. Based on this framework, we can theoretically examine the nonlinear relationship between the rapid decline rate of soil moisture and various hydrometeorological factors and identify possible flash drought risks caused by less rainfall (e.g., long dry spells), higher evapotranspiration (e.g., extreme heatwaves), lower soil water storage capacity (e.g., deforestation), or a combination thereof. Applying this framework to the global datasets, we obtained global maps of the average time for drought development and the risks of flash drought. We found that possible flash drought development in humid regions such as southern China and the northeastern United States, calling for particular attention to flash drought monitoring and mitigation.

## 1 Introduction

Drought, usually defined as a prolonged period of water scarcity, is one of the major natural disasters that influenced nearly 40% of the world population (Hamdy et al., 2003). The rapid intensification of drought is particularly detrimental such as the drought in 2012 in the central United States, which has long-term impacts on agriculture, animal husbandry, navigation, and employment (Hoerling et al., 2014), and was estimated to be the costliest drought event in U.S. history with total losses of 35 billion U.S. dollars (Grigg and Neil, 2014). The rapid intensification of drought has recently received much research attention and various indices have been proposed to define the rapid intensification of drought or 'flash drought'. Based on hydrometeorological variables such as evapotranspiration and precipitation, Mo and Lettenmaier (2015) identified two types of flash drought primarily caused by heatwaves and precipitation deficit, both of which can be accurately characterized as rapid-intensification of drought conditions (Liu et al., 2020). Moreover, soil water capacity is associated with vegetation dynamics and water balance, which acts as a buffering zone to reduce the variation of soil moisture thus also influencing the drought development (Wang et al., 2013; Gao et al., 2014; Laio et al., 2001a). While traditional drought indices and monitoring systems

(e.g., standardized precipitation evapotranspiration index) do not promptly respond to the rapid occurrence of drought events (Ford et al., 2015; Zhang et al., 2017; Mohammadi et al., 2022), soil moisture has been argued to be a useful indicator for characterizing flash drought (Hunt et al., 2009; Mozny et al., 2012; AghaKouchak et al., 2015). A flash drought event is usually identified when the pentad-average (5-day average) soil moisture dropped from a higher level (e.g., 40 percentile) to a lower one (e.g., 20 percentile) in 20 days or less (Otkin et al., 2016; Ford and Labosier, 2017; Basara et al., 2019; Nguyen et al., 2019; Lisonbee et al., 2021; Osman et al., 2021; Zhang et al., 2022) and subsequent studies have also refined the onset and end of flash drought events (Yuan et al., 2018, 2019). Readers may refer to a more comprehensive review of the flash drought definitions, for example, by Lisonbee et al. (2021).

While these indices based on pentad-average soil moisture reduce the impacts of extreme soil moisture fluctuation and are valuable for characterizing drought behaviors, the timing of soil moisture crossing any thresholds has a coarse temporal resolution of 5 days. This may be less accurate for drought occurrence within 20 days or less, resulting in a relative bias of higher than 25% and thus further complicating the assessment and identification of hydrometeorological factors contributing to the flash drought. An illustrative example based on a water balance model introduced in Sec. 2 is given in Fig. 1, which shows both the time series of instantaneous soil moisture ($s$, solid lines in Fig. 1a) and pentad-average soil moisture ($s_5$, solid lines in Fig. 1c). For the prescribed hydrometeorological factors, it takes 15 days for $s$ to decrease from 40 to 20 percentile but 15-20 days for $s_5$. When varying soil water capacity $w_0$ (Fig. 1b) or total rainfall rate by a factor of $k$ (Fig. 1d), one can find zigzag lines of $s_5$ crossing the threshold, suggesting insensitive response of traditional flash drought index to $w_0$ and $k$. While this problem may be partially solved by using a smoothing technique or changing averaging windows, it essentially stems from the probabilistic structure of the soil moisture evolution, which requires further exploration for the accurate assessment of flash drought.

Toward this goal, here we provide a stochastic framework as well as its crossing properties to quantify the rapid intensification of drought. Instead of counting drought events and justifying the proper smoothing windows for eliminating extreme fluctuations of soil moisture, we describe the whole probabilistic structure of soil moisture crossing different thresholds, which theoretically counts infinitely many drought events and smooths the extreme fluctuations over the whole spectrum of soil moisture levels. Under this framework, we can calculate the average time required for soil moisture to decline from 40 to 20 percentiles and compare the rapid decline rate of soil moisture under different hydrometeorological conditions, thus providing an efficient and objective tool for analyzing the rapid intensification of drought. The paper is organized as follows: section 2 introduces the stochastic framework, which is used to analyze various hydrometeorological factors contributing to the rapid decline rate of soil moisture and identify global patterns of flash drought risks in section 3. Section 4 discusses other factors associated with drought and the conclusions are summarized in section 5.

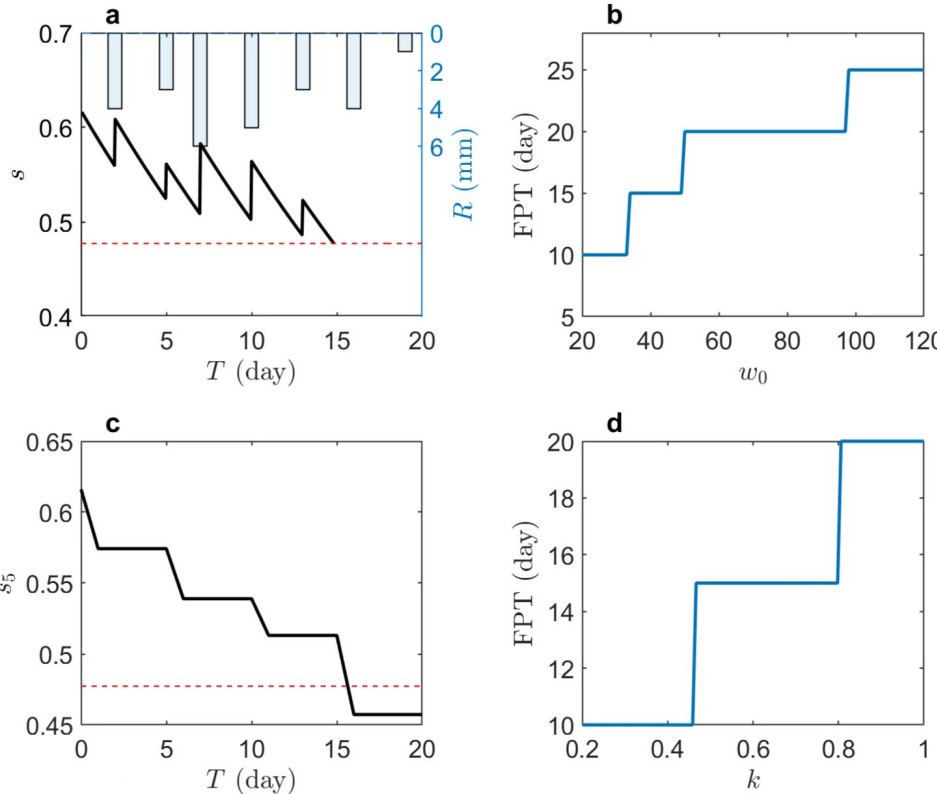

**Figure 1.** An illustrative example of drought occurrence. A water balance model in Sec. 2 is used to simulate time series of (a) rainfall, instantaneous soil moisture $s$, and (c) pentad-average soil moisture $s_5$. The first passage time (FPT) from 40 to 20 percentile is calculated (b) by varying soil water capacity $w_0$ and (d) by changing rainfall rate by a factor of $k$. The parameters in (a), (c) and (d) are given as follows: potential evapotranspiration $E_{\max} = 4\,\mathrm{mm\,day}^{-1}$ and water storage capacity $w_0 = 80$ mm.

## 2  Theory

To characterize the 'flash' behavior drought, we use, without loss of generality, the minimalist soil water balance framework (Porporato et al., 2004; Porporato and Yin, 2022)

$$w_0 \frac{\mathrm{d}x(t)}{\mathrm{d}t} = R(t) - E(x(t),t) - \mathrm{LQ}(x(t),t), \tag{1}$$

where $x$ is the relative soil moisture ranging from 0 at the wilting point to 1 around field capacity, $w_0$ is water storage capacity in the rooting zone, $R$, $E$, and LQ are rainfall, evapotranspiration, and deep leakage/runoff, respectively. In the water balance

model, $E$ is assumed to be a function of soil moisture and potential evapotranspiration, i.e.,

$$E = f\left(E_{\max}, x\right) = x E_{\max} \tag{2}$$

where the last equality assumes $E$ linearly increases from 0 for $x = 0$ to $E_{\max}$ for $x = 1$ in the minimalist framework, the excessive parts of rainfall at $x = 1$ are converted to LQ. When assuming the rainfall is a Marked Poisson process with rainfall rate $\lambda$ and exponentially distributed rainfall depth of mean $\alpha$, we can express the probability density function (PDF) of $x$ at steady state, $p(x)$, as (Porporato et al., 2004)

$$p(x) = \frac{\gamma^{\lambda/\eta}}{\Gamma(\lambda/\eta) - \Gamma(\lambda/\eta, \gamma)} e^{-\gamma x} x^{\lambda/\eta - 1} \tag{3}$$

where $\gamma = w_0/\alpha$, $\eta = E_{\max}/w_0$, and $\Gamma(\cdot)$ and $\Gamma(\cdot, \cdot)$ are the complete and incomplete gamma functions, respectively. The cumulative distribution function (CDF) of $x$ can be found by integrating Eq. (3) as

$$P(x) = \frac{\Gamma(\lambda/\eta) - \Gamma(\lambda/\eta, \gamma x)}{\Gamma(\lambda/\eta) - \Gamma(\lambda/\eta, \gamma)} \tag{4}$$

The inverse of CDF is the quantile function of $x$, providing soil moisture values for the given percentiles.

Following the flash drought definition given by Ford and Labosier (2017) and many others, we measure the timing for the drop of relative soil moisture from a high level $x_1$ (e.g., 40 percentile) to a low level $x_2$ (e.g., 20 percentile). In our stochastic framework, this timing is also a random variable, $t_{x_1 \downarrow x_2}$. While its whole distribution is difficult to obtain, its mixed feature with both continuous and discrete parts is evident (Gardiner, 1985). When there is no rainfall, the soil moisture decreases following the fast routine from $x_1$ to $x_2$, which can be found by solving Eq. (1) without rainfall and runoff (i.e., $R = \mathrm{LQ} = 0$),

$$t_{\min} = -\frac{1}{\eta} \ln\left(\frac{x_2}{x_1}\right). \tag{5}$$

The atom probability of this no rainfall condition in a Poisson process is $e^{-\lambda t_{\min}}$ (Last and Penrose, 2017). In the minimalist case, the continuous part tends to be an exponential distribution shifted by $t_{\min}$ as demonstrated in Fig. 2. Therefore, we can approximate the whole distribution of $t_{x_1 \downarrow x_2}$ as

$$f(t_{x_1 \downarrow x_2}) \approx e^{-\lambda t_{\min}} \delta(t_{x_1 \downarrow x_2} - t_{\min}) + (1 - e^{-\lambda t_{\min}}) \beta e^{-\beta(t_{x_1 \downarrow x_2} - t_{\min})}, \tag{6}$$

where $\delta(\cdot)$ is the Dirac delta function, and $\beta$ is the parameter. The cumulative probability function (CDF) of $t_{x_1 \downarrow x_2}$ can be obtained by integrating Eq. (6), i.e.,

$$F(t_{x_1 \downarrow x_2}) = \int_{t_{\min}}^{t_{x_1 \downarrow x_2}} f(\tau) d\tau = \begin{cases} 0 & t_{x_1 \downarrow x_2} < t_{\min}, \\ 1 - e^{-\beta(t_{x_1 \downarrow x_2} - t_{\min})} + e^{-\beta(t_{x_1 \downarrow x_2} - t_{\min}) - \lambda t_{\min}} & t_{x_1 \downarrow x_2} \geq t_{\min}, \end{cases} \tag{7}$$

where $\tau$ is an integration variable. This CDF can be used to quantify the risk (or probability) of first passage time lower than any given threshold. Moreover, the expectation is often referred to as the mean first passage time (MFPT), $\bar{t}_{x_1 \downarrow x_2}$ (Rodríguez-Iturbe and Porporato, 2004)

$$\bar{t}_{x_1 \downarrow x_2} = \int_{x_2}^{x_1} \frac{1}{\eta^2 u^2 p(u)} [\lambda - \lambda P(u) + \eta u p(u)] \mathrm{d}u, \tag{8}$$

which does not have an explicit solution due to the presence of $P(u)/p(u)$ in the integral. Codes for the numerical integration with different parameters are available at github.com/yxshot/MFPT. Matching this mean value with its PDF in Eq. (6) yields the parameter $\beta$

$$\beta = \frac{1 - e^{-\lambda t_{\min}}}{\bar{t}_{x_1 \downarrow x_2} - t_{\min}}. \tag{9}$$

Besides the risks given in Eq. (7), the variance of the first passage time (VFPT), $\sigma^2_{x_1 \downarrow x_2}$, could roughly quantify the uncertainties of the crossing time and can be expressed as

$$\sigma^2_{x_1 \downarrow x_2} = \int\limits_{t_{\min}}^{\infty} (t_{x_1 \downarrow x_2} - \bar{t}_{x_1 \downarrow x_2})^2 f(t_{x_1 \downarrow x_2}) dt_{x_1 \downarrow x_2}$$

$$= (t_{\min} - \bar{t}_{x_1 \downarrow x_2})^2 e^{-\lambda t_{\min}} + (1 - e^{-\lambda t_{\min}}) \left[ 2\beta^{-2} + (t_{\min} - \bar{t}_{x_1 \downarrow x_2})(t_{\min} + 2\beta^{-1} - \bar{t}_{x_1 \downarrow x_2}) \right]. \tag{10}$$

Therefore, the distributions of $t_{x_1 \downarrow x_2}$ in Eq. (6) along with its CDF in Eq. (7), mean in Eq. (8), and variance in Eq. (10) provide comprehensive metrics for quantifying the rapid intensification of drought. As a starting point for applying this framework, here we only used the minimalist model for a demonstration and a more general form of loss function for describing the risks of the flash drought will be the subject of future research (see discussion in section 4).

## 3 Results

### 3.1 Hydrometeorological impacts on the rapid decline rate of soil moisture

The interaction among climate, soil, and vegetation controls the water balance and influences drought occurrence (Mishra and Singh, 2010; Chen et al., 2021; Hu et al., 2021). This is theoretically analyzed here by using the framework developed in the last section with four hydrometeorological factors, i.e., rainfall frequency ($\lambda$), average rainfall depth ($\alpha$), potential evapotranspiration ($E_{\max}$), soil water storage capacity ($w_0$).

By fixing two factors and varying the other two, we can find how hydrometeorological factors influence the mean first passage time of soil moisture dropping from 40 to 20 percentiles. Using this stochastic framework, we found less precipitation, stronger evapotranspiration, and lower water storage capacity can speed up the loss of soil moisture, resulting in shorter MFPT (see Fig. 3a-c). While the first two factors have been identified in previous studies, the last one is less extensively investigated probably due to the low resolution of the traditional pentad-average soil moisture (although rooting depth or soil water storage capacity is one of the critical factors considered in the general drought events, e.g., Passioura, 1983; Padilla and Pugnaire, 2007; Sehgal et al., 2021). When compared with the crossing time from pentad-average soil moisture (e.g., Fig. 1), it is clear that the crossing time of the ensemble average soil moisture has smoother responses to the environmental factors, highlighting the importance of exploring the probabilistic behaviors of water balance for assessing flash drought.

Specifically, low water storage capacity accelerates the loss of water even in wet regions where plenty of water is converted into runoff (Fig. 3a) or in cold regions where potential evaporation is low (Fig. 3b). In contrast to water storage capacity, the

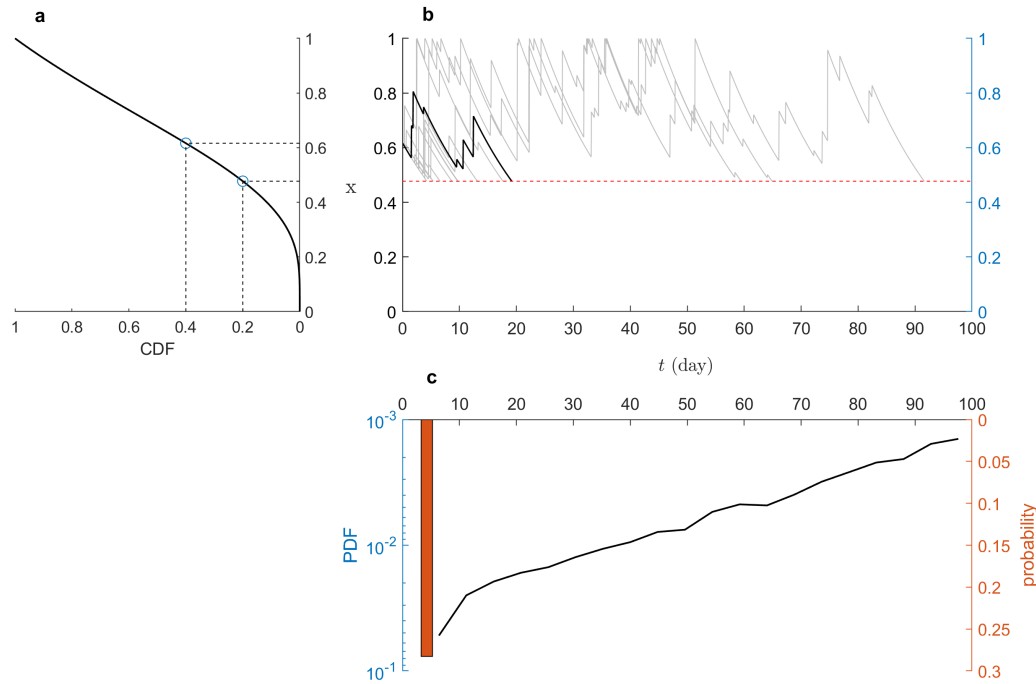

**Figure 2.** (a) Quantile function for soil moisture (i.e., the inverse of Eq.(4)) along with the 20 and 40 percentiles of soil moisture marked by circles. (b) Numerical simulation of Eq. (1) for relative soil moisture $x$ from 40 to 20 percentiles, where rainfall is assumed to be a Poisson process. (c) The empirical distribution of first passage time (sample size of 1000). The parameters are set as follows: average rainfall depth $\alpha = 12\,\text{mm}$, rainfall rate $\lambda = 0.3\,\text{day}^{-1}$, water storage capacity $w_0 = 83.2\,\text{mm}$, and potential evapotranspiration $E_{\max} = 5\,\text{mm}\,\text{day}^{-1}$. Note the continuous part of the distribution tends to be exponential (i.e., linear in the logarithmic scale for the y-axis) not only for the parameters given in this example but also in the parameter space of Fig. 2.

impacts of rainfall frequency or potential evapotranspiration on MFPT tend to be less nonlinear (Fig. 3c). In arid regions of high potential evapotranspiration rate, neither increasing water storage capacity nor rainfall rate can significantly slow down the rate of moisture decline (e.g., upper right corners of Fig. 3 b and c) due to the significant water loss. In semi-arid or semi-humid

regions, the occurrence of flash droughts may require the combined effects of several hydrological conditions (e.g., moderate rainfall frequency and high potential evapotranspiration or low water storage capacity, see Fig. 3 a and c).

Moreover, the interplay between the frequency and the depth of rainfall can be analyzed by considering a fixed total precipitation $\alpha\lambda = \text{const}$. Therefore, increasing $\lambda$ means frequent yet lighter rainfall, lowering the overall uncertainty of the rainfall process. For saturation excess runoff, lower rainfall uncertainty tends to reduce the runoff generation and thus increase the

125 MFPT as shown in Fig. 3d. Similarly, larger soil water capacity provides deeper buffering zones for uncertain rainfall, also increasing MFPT and delaying the rapid decline rate of soil moisture. Note that canopy interception is not considered here, which may reduce water infiltrated into the soil and shorten the MFPT.

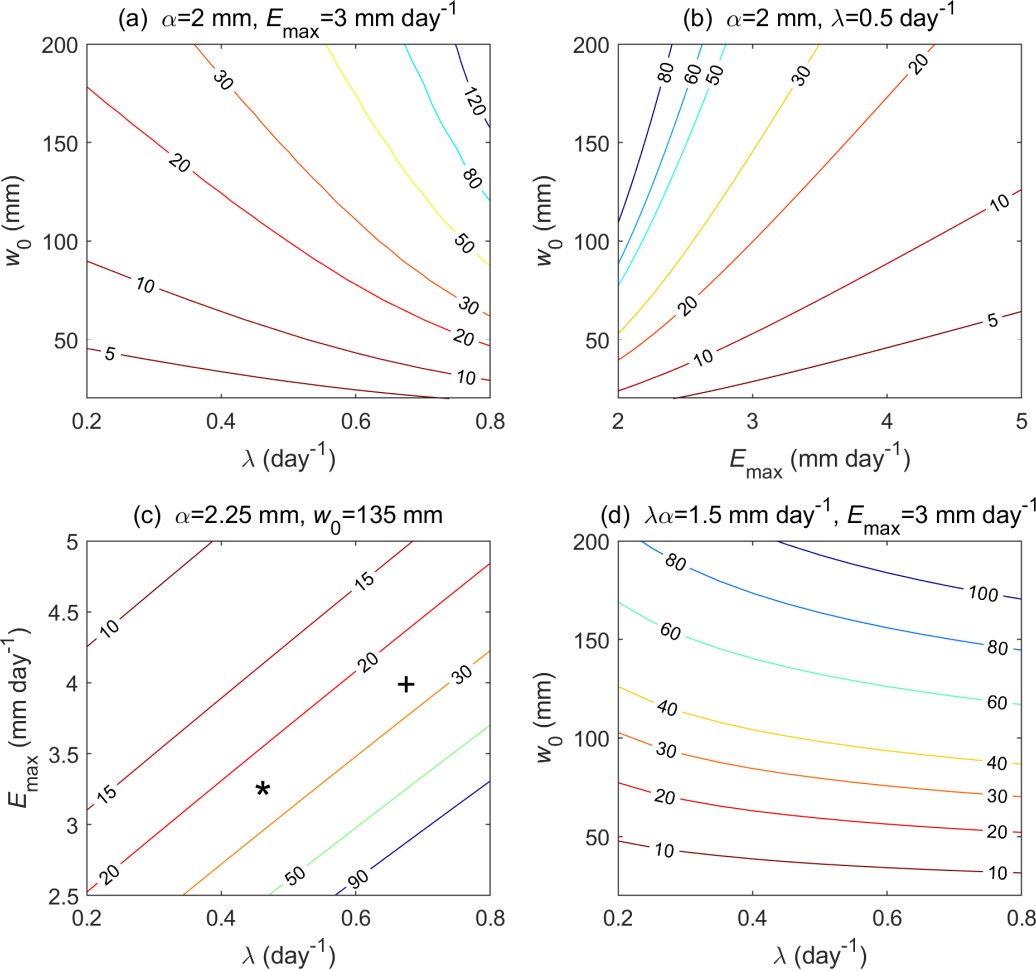

**Figure 3.** The influence of hydrometeorological factors on mean first passage time (days) of soil moisture dropping from 40 to 20 percentiles. $w_0$ is water storage capacity in the rooting zone, $\lambda$ is rainfall rate, $\alpha$ is average rainfall depth, and $E_{max}$ is potential evapotranspiration. '+' and '⋆' correspond to the for sites in Guangdong, China, and New York State, USA, the value of hydrometeorological factors and MFPT respectively (see Fig. 4).

## 3.2 Timing of global drought occurrence

Besides the theoretical analysis of the drought occurrence, our framework can also be used to diagnose the global patterns of rapid drought occurrence using global hydrometeorological datasets. The daily precipitation in the boreal summer of 2009-2018 was obtained from the Global Precipitation Climatology Project (GPCP), which combines satellite infrared, microwave, sounding observations, and precipitation observation data from more than 6,000 ground stations at the spatial resolution of $1°$ (Huffman et al., 1997, 2001). We calculated the rainfall frequency as the proportion of rainy days and rainfall depth as the average depth of daily rainfall during rainy days. We calculated the average potential evapotranspiration by using the Climate Research Unit (CRU) TS v4, which is one of the most widely used observed climate datasets at the spatial resolution of $0.5°$ (Harris et al., 2020). The global soil water storage capacity of the rooting zone was obtained from the International Satellite Land Surface Climatology Project, Initiative II (ISLSCP II) with a resolution of $1°$, which is derived from the assimilation of NDVI-fPAR and atmospheric forcing data (Kleidon, 2011).

We rescaled all these datasets to $0.5°$ spatial resolution and substituted them into Eq. (8) and Eq. (10) to find the global MFPT (see Fig. 4). We excluded hyper-arid regions, which may be better characterized as permanent drought conditions. In general, lower MFPT is located in dry and/or hot regions. It should be noted that regions with MFPT of more than 20 days are also presented, where VFPT tends to be large (see Fig. S1 in Supplementary Material). In these regions, large uncertainties of first passage time suggest flash drought is also possible due to the interannual variability of climate. More precisely, the risks of flash drought, quantified by the CDF of first passage time, were presented in Fig. (5, which show similar patterns as the global MFPT. Therefore, we only focused on MFPT in the following analysis.

Specifically, the results show that in summer soil moisture in southern China and the United States decreases rapidly, making these regions prone to flash drought risks. This is consistent with some recent observations and analyses, which have shown increasing trends of flash drought events in humid areas in China (Wang et al., 2016; Yuan et al., 2019; Qing et al., 2022). Chen et al. (2019) also found flash drought events occurred mainly in the central United States during the warm season. We focused on one site in the eastern United States and another in southeastern China, which are both marked in Fig. 3(c) according to their MFPT and hydrological conditions and in Fig. 4 based on their geographical locations. While the decline in soil moisture at both sites is around 25 days, the causes are somehow different. With approximately the same water storage capacity, the site in southeastern China has adequate precipitation but higher evaporation, whereas the site in the eastern United States has relatively lower evapotranspiration but less precipitation. These fall into the two categories of flash drought described by Mo and Lettenmaier (2015, 2016), namely heat-wave flash drought caused by increased evapotranspiration and the precipitation-deficit flash drought caused by insufficient precipitation. From our stochastic framework, it might be interesting to define a third type of flash drought related to low water capacity in regions undergoing rapid urbanization or deforestation. This requires further investigation and remains an exciting and open area of research in hydrometeorology.

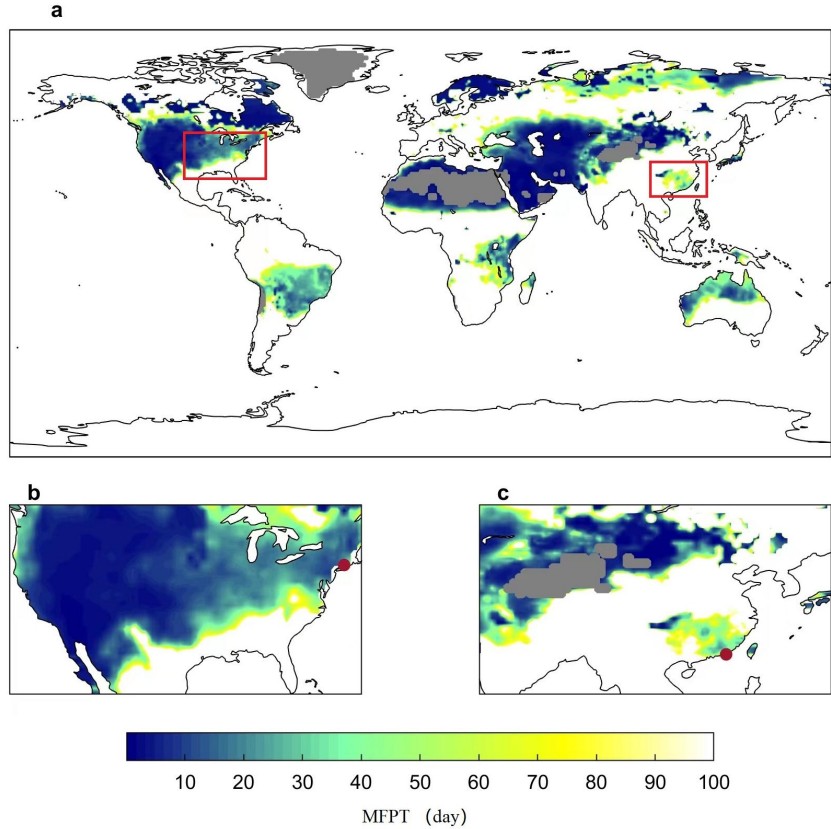

**Figure 4.** Global distribution of mean first passage time (MFPT) in summer. The red boxes in panel (a) indicate the zoomed areas in panel (b) and (c), where New York State, USA (74°W, 44°N) and Heyuan City, Guangdong Province, China (115°E, 24°N) are marked by red dots. The gray areas are hyper-arid regions, other colored areas are those where the MFPT of soil moisture drops from 40 to 20 percentiles in less than 100 days. Desert regions (grey areas) are excluded from this analysis.

## 4 Discussion

We have provided a stochastic framework to quantify the timing of soil moisture crossing from one level to another to characterize the occurrence of flash drought. While conventionally pentad-average soil moisture has been used to estimate the crossing properties, the soil moisture at daily timescale in our stochastic framework is not directly used to characterize the flash drought. Instead, the ensemble averages of the first passage time (i.e., averaged over many realizations of the stochastic processes) are much smoother than the first passage time for the given hydrometeorological condition and are used to characterize the rapid intensification of drought. The crossing properties of the pentad-average soil moisture should asymptotically approach the MFPT, which could provide a more accurate description of the soil moisture dry-down process.

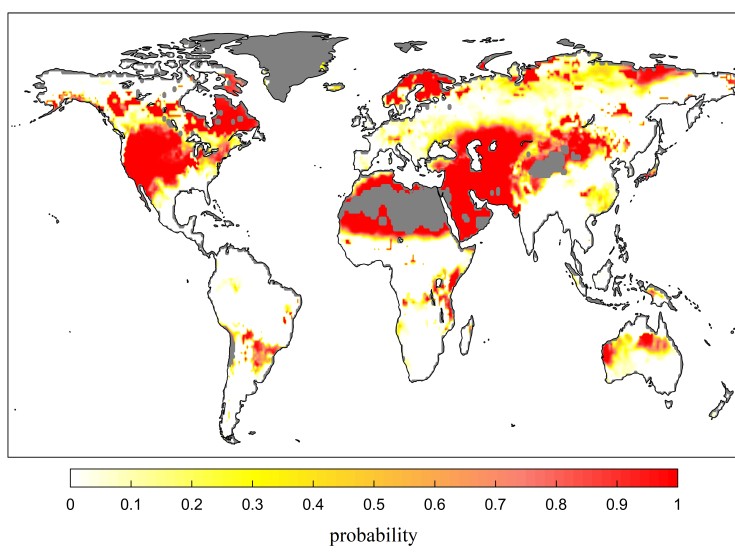

**Figure 5.** Global risk of flash drought occurrence. Risk is calculated from Eq. (7) as the probability of soil moisture dropping from 40 to 20 percentiles within 20 days or less. Similar patterns can be found by using different thresholds.

Besides the soil moisture, evaporative stress ratios ($E/E_{\max}$) or evaporation deficit ($E - E_{\max}$x) were also used to characterize flash droughts (e.g., Li et al., 2020; Christian et al., 2021). In the minimalist framework with $E = xE_{\max}$, the evaporative stress ratio is already equivalent to $x$. In the more general form, when modeling evaporation as a function of soil potential evaporation and soil moisture, we can model evapotranspiration with different soil water thresholds (e.g., wilting point and field capacity) and still obtain the statistics of crossing properties (Rodríguez-Iturbe and Porporato, 2004). In this general framework, these two new metrics can be expressed as functions of $E_{\max}$ and $x$. If daily variations of $E_{\max}$ were assumed to have limited impacts on soil water balance (e.g., Daly and Porporato, 2006), these metrics can be expressed as the derived distributions of soil moisture, allowing us to analyze flash drought using the corresponding percentiles and crossing properties of evaporative stress ratios or deficit.

In the minimalist model, drought was explicitly diagnosed with only four hydrometeorological parameters. This analysis, however, can be extended to explore other important factors. In this regard, we specifically consider the impacts of deforestation and heatwaves. Deforestation converts forests into cropland or savanna, possibly reducing the rooting depth and soil water storage capacity (Kleidon and Heimann, 1999; Nijzink et al., 2016; O'Connor et al., 2019). As shown in Fig. 3, lower soil water storage capacity ($w_0$) tends to reduce the mean first passage time of soil moisture dropping from 40 to 20 percentiles, demonstrating the possible impacts of deforestation on flash drought.

Moreover, deforestation also tends to increase surface albedo and thus influences the surface energy balance and potential evaporation rate (Dirmeyer and Shukla, 1994; Cerasoli et al., 2021). Smaller $E_{\max}$ increases the mean first passage time and therefore reduces the likelihood of flash drought (see Fig. 3 b and c). Deforestation may also change soil properties such as

organic content, retention curve, and infiltration rate (Runyan et al., 2012; Veldkamp et al., 2020), which inevitably influence the hydrological cycle and soil moisture dynamics (Laio et al., 2001b). Such changes can be included in the full stochastic framework (e.g., Rodríguez-Iturbe and Porporato, 2004) to diagnose the indirect impacts of deforestation on flash drought.

At an even large scale, deforestation may also change surface temperature and precipitation through land-atmosphere interaction (Shukla et al., 1990; Salazar et al., 2016). Deforestation may change the partitioning of surface heat flux and influence the atmospheric boundary layer dynamics, controlling the transition from shallow to deep convection (Betts et al., 1996; Findell and Eltahir, 2003; Yin et al., 2015; Tuttle and Salvucci, 2016; Cerasoli et al., 2021). A lower precipitation rate corresponds to a faster drop of soil moisture and a higher probability of flash drought as shown in Fig. 3 a and c).

As one of the important contributors to flash drought, heatwaves are often accompanied by high temperatures and strong solar radiation Stott et al. (2004). From Penman equation (see Eq. (1) in supplementary material), we expect higher equilibrium evaporation and larger $E_{\max}$. Moreover, dry or moist heatwaves may also have abnormal vapor pressure deficit (Stefanon et al., 2012), which may influence the drying power of the air and also $E_{\max}$. Therefore, heatwaves could control the soil moisture dynamics and drought occurrence by changing the potential evapotranspiration.

## 5   Conclusions

We have used a stochastic framework to quantify the rapid intensification of drought. Within the minimalist soil water balance framework, we provided the mean first passage time for the relative soil moisture dropping from different levels, which was then used to identify different types of flash drought. We found that not only precipitation and evapotranspiration frequently mentioned in previous studies but also water storage capacity discussed here could all play major roles in controlling the rapid decline rate of soil moisture. By applying this framework and analyzing various hydrometeorological factors, we identified a rapid decline of soil moisture in some wet areas due to high evapotranspiration rates, such as southern China and the northeast United States.

In response to global warming, the frequency of flash droughts may increase, posing great risks to our society. Understanding the causes of these drought events is a necessary step for drought warning, preparation, and mitigation. The stochastic framework developed here is efficient at diagnosing the impacts of hydrometeorological factors and thus could provide an objective tool for monitoring flash drought events. Future work could focus on applying this stochastic framework and using the upcrossing properties of the stochastic process to evaluate the drought-mitigation strategies by quantifying the timing of recovering from a low soil moisture level to a higher level (e.g., setting $x_1 < x_2$ for $\bar{t}_{x_1 \uparrow x_2}$).

*Code availability.*   Codes for calculating MFPT are available at github.com/yxshot/MFPT.

*Data availability.* GPCP data were obtained from doi.org/10.7289/V5RX998Z, CRU data were from crudata.uea.ac.uk/cru/data/hrg/, ISLSCP II data were from daac.ornl.gov/ISLSCP_II/.

*Acknowledgements.* J.Y. acknowledges support from Natural Science Foundation of Jiangsu Province (BK20221343), the National Natural Science Foundation of China (41877158, 51739009), NUIST startup funding (1441052001003), and NUIST's supercomputing center.

*Author contributions.* YL SC JY and XY conceived and designed the study. YL wrote an initial draft of the paper, to which all authors contributed edits throughout.

*Competing interests.* The authors declare no competing interests.

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
