# Peer review of "Technical note: A stochastic framework for identification and evaluation of flash drought"

_Hydrology and Earth System Sciences, 2022_

## Referee Comment (RC1)

**Review of the manuscript:**

***Technical note: A stochastic framework for identification and evaluation of flash drought***

Anonymous Reviewer #1

Li, Chen, Yin and Yuan(2022) propose a new approach for flash drought identification and characterization based on a simple water balance model for tracking soil moisture changes. The mathematical and probabilistic framework is sound and the results derived from it provide robust mathematical evidence to current understanding of flash drought development. The method, despite the limitation of parameterization, allows the simulation of multiple scenarios to assess different types of flash droughts.

**General comments**

The manuscript is written in good English and overall is well structured. The authors provide extensive literature review and good methodological description

- Please improve your code documentation and comment.

It is my opinion that the manuscript has no major technical flaws. Nevertheless, our recommendation is for *Minor Reviews*.

**Specific comments**

1. Results

- Fig 4:

    o Please improve the colour scheme, as the points in New York and Heyuan are barely visible.

    o Why did you use a limit of 100 days in the scale? Normally flash droughts intensification period is limited to up to 30 days (Osman et al, 2021; Ford and Laosier, 2015, Lisonbee et al, 2021).

    o By using the metric of Mean First Passage Time (MFPT), some areas end up showing no actual flash droughts. Please consider showing the 10th percentile of first passage time, that would show the expected occurrence in more areas.

    o Please justify the very low MFPT in semi-arid regions, such as southern India, northern Namibia/Botswana and northeast Brazil

---

## Community Comment (CC1)

**DEPARTMENT OF HYDROMETEOROLOGY**
NANJING UNIVERSITY OF INFORMATION SCIENCE & TECHNOLOGY,
NANJING, JIANGSU, CHINA, 210044

JUN YIN
PROFESSOR
DEPARTMENT OF HYDROMETEOROLOGY

TEL +86-25-58731556
EMAIL: JUN.YIN@NUIST.EDU.CN

Nanjing, Oct 25, 2022

Dear Reviewer,

We would like to thank you for the careful and constructive assessment of our manuscript (hess-2022-313). We are also grateful to you for the insightful and valuable comments. Following these suggestions, we have revised the manuscript as detailed in the next pages. We thank you and also welcome your further questions and comments.

Sincerely,

**Reviewer comment (italicized) is followed by a response.**

*General comments:*

*The manuscript is written in good English and overall is well structured. The authors provide extensive literature review and good methodological description*

We thank the reviewer for the positive comments and encouragement.

*• Please improve your code documentation and comment.*

The code documentation has been revised. A screenshot of the documentation is reported below and further information can be accessed from "github.com/yxshot/MFPT".

[Figure]

**Figure 1** The screenshot of the documentation.

*Fig 4. Please improve the colour scheme, as the points in New York and Heyuan are barely visible.*

Thank you for pointing this out. We used a new color scheme and also provided two insets for New York and Heyuan. The new map will be updated in the revised manuscript and is reported below.

[Figure]

**Figure 2** Global distribution of mean first passage time (MFPT) in summer. The two points marked in red are New York State, USA and Heyuan City, China. The gray areas are hyper-arid regions, other colored areas are those where MFPT of soil moisture dropping from 40 to 20 percentiles in less than 100 days. Desert regions (grey areas) are excluded in this analysis.

Good point. We cited these references to clarify that flash drought is usually up to 30 days.

However, it is still possible to have flash drought for large MFPT, which only tells the long-term averages of the intensification period. To address this point, we also derived the variance of the first passage time (VFPT) as

$$\sigma^2_{x_1\downarrow x_2} = \int_{t_{\min}}^{\infty} (t_{x_1\downarrow x_2} - \bar{t}_{x_1\downarrow x_2})^2 f(t_{x_1\downarrow x_2}) dt_{x_1\downarrow x_2}$$

$$= (t_{\min} - \bar{t}_{x_1\downarrow x_2})^2 e^{-\lambda t_{\min}} + (1 - e^{-\lambda t_{\min}}) \left[ 2\beta^{-2} + (t_{\min} - \bar{t}_{x_1\downarrow x_2})(t_{\min} + 2\beta^{-1} - \bar{t}_{x_1\downarrow x_2}) \right].$$

As shown in Figure 3 below, VFPT has similar spatial patterns as MFPT. The limit is set to 100 days in order to show as much as possible the areas where flash droughts are likely to occur. We will incorporate these results in the revised manuscript.

[Figure]

**Figure 3** Global distribution of variance of the mean first passage time (VFPT) (units: day$^2$).

*Fig 4. By using the metric of Mean First Passage Time (MFPT), some areas end up showing no actual flash droughts. Please consider showing the 10th percentile of first passage time, that would show the expected occurrence in more areas.*

Thank you for the suggestion.

Similar to the VFPT as shown in Fig. 3, the 10th percentile suggested by the reviewer could also be a useful indicator to quantify the uncertainties of drought intensification period. However, there is one technique problem regarding to the atom probability of this first passage time. As shown Fig. 2c in the manuscript and also reported below in Fig. 4, the whole distribution is neither continuous nor discrete but a mixed one. The atom probability is associated with the condition where there is hardly any rainfall before the soil moisture drops to the threshold. It is possible that 10th percentile just fall into this atom probability and end up as $t_{min}$. To provide more accurate quantification of drought intensification period, we will include both MFPT and VFPT maps in the revised manuscript.

[Figure]

**Figure 4** The corresponding distribution of first passage time (sample size of 1000).

*Fig 4. Please justify the very low MFPT in semi-arid regions, such as southern India, northern Namibia/Botswana and northeast Brazil.*

We are not quite sure if we understand this comment correctly.

As shown in Figure 2 above, the MFPT in southern India, northern Namibia/Botswana and northeast Brazil are quite high (i.e., close to or higher than 100 days, see). Aside from climate conditions, the long crossing time in these regions may be associated with deep rooting depths (see root-zone storage, $w_0$, in Figure 5 below), which acts as a buffering zone to reduce the variation of soil moisture and thus increase the time for drought intensification (e.g., Laio et al. 2001). We will clarify the role of rooting depths in the revised manuscript.

Please feel free to correct us if we misunderstood your comment. Thank you again!

[Figure]

**Figure 5** Global distribution of soil water storage capacity (units: mm), which used to calculate global MFPT.

**References**

Laio, F., A. Porporato, C. P. Fernandez-Illescas, and I. Rodriguez-Iturbe, 2001: Plants in water-controlled ecosystems: active role in hydrologic processes and response to water stress IV. Discussion of real cases. Adv. Water Resour., 18.

---

## Author Comment (AC1)

**DEPARTMENT OF HYDROMETEOROLOGY**
NANJING UNIVERSITY OF INFORMATION SCIENCE & TECHNOLOGY,
NANJING, JIANGSU, CHINA, 210044

JUN YIN
PROFESSOR
DEPARTMENT OF HYDROMETEOROLOGY

TEL +86-25-58731556
EMAIL: JUN.YIN@NUIST.EDU.CN

Nanjing, Oct 26, 2022

Dear Reviewer,

We would like to thank you for the careful and constructive assessment of our manuscript (hess-2022-313). We are also grateful to you for your insightful and valuable comments. Following these suggestions, we have revised the manuscript as detailed in the next pages. We thank you and also welcome your further questions and comments.

Sincerely,

**Reviewer comment (italicized) is followed by a response.**

*General comments:*

*The manuscript is written in good English and overall is well structured. The authors provide extensive literature review and good methodological description*

We thank the reviewer for the positive comments and encouragement.

*• Please improve your code documentation and comment.*

The code documentation has been revised. A screenshot of the documentation is reported below and further information can be accessed from "github.com/yxshot/MFPT".

[Figure]

**Figure 1** The screenshot of the documentation.

*It is my opinion that the manuscript has no major technical flaws. Nevertheless, our recommendation is for Minor Reviews.*

*Specific comments:*

*Fig 4. Please improve the colour scheme, as the points in New York and Heyuan are barely visible.*

Thank you for pointing this out. We used a new color scheme and also provided two insets for New York and Heyuan. The new map will be updated in the revised manuscript and is reported below.

[Figure]

**Figure 2** Global distribution of mean first passage time (MFPT) in summer. The two points marked in red are New York State, USA, and Heyuan City, China. The gray areas are hyper-arid regions, other colored areas are those where the MFPT of soil moisture drops from 40 to 20 percentiles in less than 100 days. Desert regions (grey areas) are excluded from this analysis.

*Fig 4. Why did you use a limit of 100 days in the scale? Normally flash droughts intensification period is limited to up to 30 days (Osman et al, 2021; Ford and Laosier, 2015, Lisonbee et al, 2021).*

Good point. We cited these references to clarify that flash drought is usually up to 30 days.

However, it is still possible to have a flash drought for large MFPT, which only tells the long-term averages of the intensification period. To address this point, we also derived the variance of the first passage time (VFPT) as

$$\sigma^2_{x_1\downarrow x_2} = \int_{t_{\min}}^{\infty} (t_{x_1\downarrow x_2} - \bar{t}_{x_1\downarrow x_2})^2 f(t_{x_1\downarrow x_2}) dt_{x_1\downarrow x_2}$$
$$= (t_{\min} - \bar{t}_{x_1\downarrow x_2})^2 e^{-\lambda t_{\min}} + (1 - e^{-\lambda t_{\min}}) \left[ 2\beta^{-2} + (t_{\min} - \bar{t}_{x_1\downarrow x_2})(t_{\min} + 2\beta^{-1} - \bar{t}_{x_1\downarrow x_2}) \right].$$

As shown in Figure 3 below, VFPT has similar spatial patterns as MFPT. The limit is set to 100 days in order to show as much as possible the areas where flash droughts are likely to occur. We will incorporate these results in the revised manuscript.

[Figure]

**Figure 3** Global distribution of the variance of the mean first passage time (VFPT) (units: day$^2$).

*Fig 4. By using the metric of Mean First Passage Time (MFPT), some areas end up showing no actual flash droughts. Please consider showing the 10th percentile of first passage time, that would show the expected occurrence in more areas.*

Thank you for the suggestion.

Similar to the VFPT as shown in Fig. 3, the 10th percentile suggested by the reviewer could also be a useful indicator to quantify the uncertainties of the drought intensification period. However, there is one technical problem regarding the atom probability of this first passage time. As shown in Fig. 2c in the manuscript and also reported below in Fig. 4, the whole distribution is neither continuous nor discrete but a mixed one. The atom probability is associated with the condition where there is hardly any rainfall before the soil moisture drops to the threshold. It is possible that the 10th percentile just falls into this atom probability and ends up as $t_{min}$. To provide a more accurate quantification of the drought intensification period, we will include both MFPT and VFPT maps in the revised manuscript.

[Figure]

**Figure 4** The corresponding distribution of first passage time (sample size of 1000).

*Fig 4. Please justify the very low MFPT in semi-arid regions, such as southern India, northern Namibia/Botswana and northeast Brazil.*

We are not quite sure if we understand this comment correctly.

As shown in Figure 2 above, the MFPT in southern India, northern Namibia/Botswana, and northeast Brazil are quite high (i.e., close to or higher than 100 days, see). Aside from climate conditions, the

long crossing time in these regions may be associated with deep rooting depths (see root-zone storage, $w_0$, in Figure 5 below), which acts as a buffering zone to reduce the variation of soil moisture and thus increase the time for drought intensification (e.g., Laio et al. 2001). We will clarify the role of rooting depths in the revised manuscript.

Please feel free to correct us if we misunderstood your comment. Thank you again!

[Figure]

**Figure 5** Global distribution of soil water storage capacity (units: mm), which is used to calculate global MFPT.

**References**

Laio, F., A. Porporato, C. P. Fernandez-Illescas, and I. Rodriguez-Iturbe, 2001: Plants in water-controlled ecosystems: active role in hydrologic processes and response to water stress IV. Discussion of real cases. Adv. Water Resour., 18.

---

## Author Comment (AC3)

**DEPARTMENT OF HYDROMETEOROLOGY**
NANJING UNIVERSITY OF INFORMATION SCIENCE & TECHNOLOGY,
NANJING, JIANGSU, CHINA, 210044

JUN YIN
PROFESSOR
DEPARTMENT OF HYDROMETEOROLOGY

TEL +86-25-58731556
EMAIL: JUN.YIN@NUIST.EDU.CN

Nanjing, Dec 26, 2022

Dear Reviewer,

We thank you for reviewing our manuscript (hess-2022-313) and for your positive and constructive comments. These comments are all valuable and helpful for improving our article. In particular, we made the following main changes, motivated by your comments:

- We discussed the impacts of deforestation on soil water storage, evaporation, and rainfall to clarify the linkages between deforestation and flash drought.
- We explained the temporal averages essentially approach the ensemble averages to justify the use of daily timescale in the stochastic water balance model.
- We clarified the differences between evapotranspiration and potential evapotranspiration.

Detailed responses to your comments and suggestions were reported in the next pages.

Sincerely,

**Reviewer comment (italicized) is followed by a response.**

*General comment*

*This study used a stochastic water balance framework to examine the nonlinear relationship between the timing of drought and various hydrometeorological factors and identify possible flash drought events caused by lack of rainfall, high evapotranspiration, low soil water storage capacity, or a combination thereof. Indeed, there are a variety of definitions for flash drought, which has been merged as a critical sub-seasonal phenomenon with great impacts on agriculture, the economy, and society. Providing new metrics for flash drought from a stochastic perspective is certainly of great importance to our understanding of the rapid intensification of drought events. The stochastic theory is sound and straightforward, and the authors also found that flash drought also exists in humid regions such as southern China and the northeastern United States, calling for particular attention to flash drought monitoring and mitigation. And the manuscript is wellwritten and well structured, with potential publication in HESS. Below I list some points and the authors are wished to address before published.*

We are grateful to the reviewers for their positive comments and encouragement. We have used these suggestions to improve the manuscript.

*Major concerns*

*As illustrated in the text, the proposed framework measures the effect of deforestation on flash drought, but the description on this content is unclear. Soil water storage capacity does have a strong link with vegetation distribution, for example, drylands, with low NDVI, correspondingly show weak soil water storage capacity. In addition, deforestation can change hydrological and energy cycle processes, such as altering surface albedo and soil infiltration rate, which have an impact on flash drought. What is the relationship between deforestation and soil water storage capacity? Please add some specific statements. Further explaining is also needed, from my viewpoint, on how the framework measures the effect of deforestation on flash drought.*

Thank you for pointing this out. We did not explain this very well in the original manuscript, but now we have clarified the linkage between deforestation and flash drought.

As commented by the reviewer, deforestation converts forests into cropland or savanna, possibly reducing the rooting depth and soil water storage capacity (Kleidon and Heimann 1999; O'Connor et al. 2019; Nijzink et al. 2016). As shown in Figure 1 a and b (Fig. 3 in the manuscript), lower soil water storage capacity ($w_0$) tends to reduce the mean first passage time of soil moisture dropping from 40 to 20 percentiles, demonstrating the possible impacts of deforestation on flash drought.

Moreover, deforestation also tends to increase surface albedo and thus influence the surface energy balance and potential evaporation rate (Dirmeyer and Shukla 1994; Cerasoli et al. 2021), which have been considered in the stochastic framework. Smaller $E_{max}$ increases the mean first passage time and therefore reduces the likelihood of flash drought (see Figure 1 b and c).

The changes in soil properties after deforestation have been reviewed by Runyan et al. (2012) and Veldkamp et al. (2020) and many others. Such changes in soil organic content, retention curve, and infiltration rate inevitably influence the hydrological cycle and soil moisture dynamics (Laio et al. 2001). It is possible to include all these factors in the full stochastic framework (e.g., Rodríguez-Iturbe and Porporato 2004) to diagnose the impacts of deforestation on soil properties and the rapid decline rates of soil moisture.

At an even large scale, deforestation may also change surface temperature and precipitation through land-atmosphere interaction (Shukla et al. 1990; Salazar et al. 2016). Deforestation may change the partitioning of surface heat flux and influence the atmospheric boundary layer dynamics, controlling the transition from shallow to deep convection (Betts et al. 1996; Findell and Eltahir 2003; Yin et al. 2015; Tuttle and Salvucci 2016; Cerasoli et al. 2021). Lower precipitation rate corresponds to a faster drop of soil moisture and a higher probability of flash drought as shown in Figure 1 a and c.

We will discuss all these linkages between deforestation and flash drought in the revised manuscript.

[Figure]

Figure 1  The influence of hydrometeorological factors on mean first passage time (days) of soil moisture dropping from 40 to 20 percentiles.

*Existing model simulations or satellite observations can provide daily-scale soil moisture as well, although these data are not free from biases. In comparison to traditional droughts, flash droughts are characterized by rapid development, while the rapid development of flash droughts usually occurs within days or weeks, so pentadscale hydrometeorological variables are commonly used and few studies analyzed flash droughts based on daily-scale data. The necessity to study the timing of flash drought based on the minimalist hydrological model should be further explained and discussed.*

Good point. Actually, we already did this, but we did not explain this approach very well in the previous version of the manuscript.

As commented by the reviewer, flash drought is often characterized by the pentad (5-day) average soil moisture, which may have smoother temporal evolution than the daily soil moisture. While soil moisture is modeled at a daily timescale in our stochastic framework (see gray and black lines in Figure 2 top panel), the corresponding time for soil moisture dropping from 40 to 20 percentiles (first passage time, see the distribution in Figure 2 bottom panel) is NOT directly used to characterize the flash drought. Instead, the ensemble averages of the first passage time (i.e., averaged over many realizations of the stochastic processes) are much smoother than the first passage time for the given hydrometeorological condition and are used to characterize the rapid intensification of drought.

In fact, the soil moisture averaged over a long period is equivalent to the ensemble average under the ergodic hypothesis, which is usually valid in a chaotic system such as the soil water dynamics driven by stochastic forcing (Eckmann and Ruelle 1985; Duan et al. 2002) at the scales considered here. In its strictest form, the ergodic hypothesis states that ensemble statistics at any given time or position are identical to the temporal or spatial statistics (mean and higher-order moments). Therefore, the crossing time of the pentad average soil moisture should asymptotically approach to the MFPT used in this study, which could provide an accurate description of the soil moisture dry-down process.

In the revised manuscript, we will clarify the differences between the first passage time and the mean first passage time and explicitly state that the latter is used to characterize the flash drought.

[Figure]

Figure 2 (top) numerical simulation of water balance for relative soil moisture x dropping from 40 to 20 percentiles, and (bottom) the corresponding distribution of first passage time (sample size of 1000).

*One more point I concern is that the framework can measure the effect of evapotranspiration (E) on flash drought, yet there is difference between potential evapotranspiration (PET) and E, for example for moisture-limited dry lands. I don't know did the authors measure the difference between E and PET on the results in Figure 3? In addition, the change in E is related to heatwave, while other factors (such as change in leaf area index and solar radiation) can also impact E. I suggest adding some discussion, in particular, on the difference between E and PET.*

Thank you for the valuable suggestions.

Yes, the differences between $E$ and PET were considered in Figure 3. While these differences have been briefly discussed in the original manuscript (Line 58), it is not very clear and will be explicitly addressed in the revised manuscript.

In the water balance model, $E$ is assumed to be a function of soil moisture and potential evapotranspiration, i.e.,

$$E = f(E_{max}, x) = xE_{max} , \tag{1}$$

where the last equality assumes $E$ linearly increases from 0 for $x = 0$ to $E_{max}$ for $x = 1$ in the minimalist framework. It should be noted that a more general form of $f(E_{max}, x)$ can still be solved analytically for the mean first passage time. Therefore, we can model evapotranspiration with different soil water thresholds such as the wilting point, onset of the soil water stress, and field capacity in the more general stochastic framework to explore the mean first passage time and the flash drought.

Moreover, as commented by the reviewer, heatwave, leaf area index, and solar radiation also influence flash drought, which will be thoroughly discussed in the revised manuscript. Specifically, we will use the Penman equation to introduce the potential evapotranspiration as

$$E_{max} = \underbrace{\frac{\Delta}{\rho_w \lambda_w (\Delta + \gamma)} Q}_{E_e} + \underbrace{\frac{\gamma}{\rho_w (\Delta + \gamma)} \left( \frac{\varepsilon}{p_0} \rho g_a \text{VPD} \right)}_{E_v}, \tag{2}$$

where $E_e$ is equilibrium evapotranspiration, $E_v$ is the evapotranspiration due to the drying power of the air, $\Delta$ is the slope of the saturation vapor pressure curve (a nonlinear function of air temperature), $\gamma$ is psychrometric constant, $\lambda_w$ is the latent heat of water vaporization, $Q$ is available surface energy, $\varepsilon$ is the ratio of the gas constant for dry air to that of water vapor, $p_0$ is near-surface air pressure, $\rho$ is the air density, $\rho_w$ water density, $g_a$ is aerodynamic conductance, and VPD is vapor pressure deficit. Heatwave is often accompanied by high temperature and strong solar radiation (Stott et al. 2004), which tend to increase $E_e$; dry or moist heatwaves may also have abnormal VPD (Stefanon et al. 2012), which may influence $E_v$. Vegetation with a larger leaf area index tends to have higher surface roughness, resulting in larger $g_a$ and $E_v$. Therefore, heatwave, leaf area index, and solar radiation influence the potential evapotranspiration, which will further control the soil moisture dynamics and drought occurrence.

We will incorporate these discussions in the revised manuscript.

*Minor concerns*

*Aside from soil moisture, evaporation deficit (PET-ET) or evaporative stress ratio (ET/PET) is often closely monitored to quantify the intensification of flash drought. It would be useful also to provide a more general framework to consider these variables (or at least these variables should be acknowledged).*

These are valid points; thank you!

As commented by the reviewer, evaporative stress ratios ($E/E_{max}$) or evaporation deficit ($E-E_{max}$) were also used to characterize flash droughts (e.g., Li et al. 2020; Christian et al. 2021). In the minimalist framework with $E = xE_{max}$, the evaporative stress ratio is already equivalent to $x$, which has been discussed in the original manuscript. In the more general form, when modeling evaporation as a function of soil potential evaporation and soil moisture, we can further rewrite these two metrics as

$$\begin{cases} \dfrac{E}{E_{max}} = \dfrac{f(E_{max},x)}{E_{max}} \\ E - E_{max} = f(E_{max},x) - E_{max} \end{cases}, \quad (3)$$

which are functions of $E_{max}$ and $x$. If we assume the daily variations of $E_{max}$ have limited impacts on soil water balance (Daly and Porporato 2006), we can treat evaporative stress ratios or deficit as the derived distributions of soil moisture, allowing us to link the corresponding percentiles and crossing properties to these for soil moisture.

In the revised manuscript, we will discuss these metrics in the stochastic framework for diagnosing flash drought.

*Line 75: the example given in Fig. 2c clearly shows an exponential tail. Can we still have exponential distribution for parameters with different values? This should be explored.*

We still have exponential tails, which were explored by linearly fitting the logarithmic of the tails for different parameters. From our numerical simulations ($n = 1000$) in the parameter space of Figure 1, the correlation coefficients between $\log(p(t_{x_1 \downarrow x_2}))$ and $t_{x_1 \downarrow x_2}$ for $t > t_{min}$ are close to -1 with mean value of -0.94 and standard deviation of 0.03, suggesting these tails could be properly described as exponential. These results will be reported in the revised manuscript.

*Abstract should also emphasize the probabilistic structure of the first passage time, which is the benefit of the stochastic framework.*

Thank you for your suggestion. Particularly, we stressed that the stochastic water balance framework can be used to describe the probability of the timing for soil moisture dropping from a higher level to a lower one.

*Line 13: period is missed after the citation.*

Corrected.

*I think there should be minus sign in front of Eq. (4).*

Thank you for your reminder.

*Line 74: The atom probability of no rainfall is not trivial. Please provide references or details of its derivation.*

The reference of Last and Penrose (2017) will be provided in the revised manuscript.

*How to calculate the rainfall frequency and average depth. Please clarify.*

The time series of the daily precipitation in the boreal summer of 2009-2018 was obtained from the Global Precipitation Climatology Project (GPCP). From these records, we calculated the rainfall frequency as the proportion of rainy days and rainfall depth as the average daily rainfall depth (excluding days without rainfall). This will be clarified at the beginning of Sec. 3.2.

*The information provided by each picture is seemingly not enough. Can you add more information, please?*

Thank you for the reminder.
In the revised manuscript, we will provide more information in the captions to specify all parameters and data sources used in the figure.

*Discussion chapters should be added to enrich the content*

Thank you for pushing us to improve the presentation of the manuscript. In the revised reversion, we will add a discussion section to address the potential impacts of heatwave, deforestation, leaf area index, solar radiation, and land-atmosphere interaction on the flash drought. We will also include the Penman equation in the Appendix to help explain that larger potential evapotranspiration will reduce the time for soil moisture to drop from a higher level to a lower one.

*I think the explanation of "timing of drought" in the text is slightly vague, which may further affect the readers' understanding of the drought risk mentioned in the study. Please add some explanations for this concept.*

To be consistent with previous studies (e.g., Li et al. 2020; Christian et al. 2021), we will use the 'rapid decline rate of soil moisture' to characterize the flash drought throughout the text.

*Could you point out the numerical interval of timing of drought with high risk of flash drought?*

This is an interesting question. Thank you!
As suggested in previous studies (e.g., Otkin et al., 2016; Ford and Labosier, 2017; Basara et al., 2019; Nguyen et al., 2019; Zhang et al., 2022), the flash drought is often characterized by soil moisture dropping from 40 to 20 percentiles within 20 days. Therefore, the risk of flash drought can be quantified as the probability of first passage time lower than 20 days (or any other threshold), which is exactly the cumulative probability function (CDF). In our stochastic framework, CDF can be obtained by integrating Eq. (5), i.e.,

$$F(t_{x_1 \downarrow x_2}) = \int_{t_{\min}}^{t_{x_1 \downarrow x_2}} f(\tau) d\tau = \begin{cases} 0 & t_{x_1 \downarrow x_2} < t_{\min} \\ 1 - e^{-\beta(t_{x_1 \downarrow x_2} - t_{\min})} + e^{-\beta(t_{x_1 \downarrow x_2} - t_{\min}) - \lambda t_{\min}} & t_{x_1 \downarrow x_2} \geq t_{\min} \end{cases} \tag{4}$$

Using this equation, we can calculate the global risk of flash drought as shown in Figure 3, which has similar patterns as the global MFPT (i.e., Figure 4 in the original manuscript). Note that risk with different thresholds (e.g., 25 or 30 days) can still be obtained from Eq. (4).

By inversing Eq. (4), we can also obtain the thresholds for any given risk (i.e., probability). In the revised manuscript, we will add this global risk figure to explain the unique feature of our methods.

We thank you again for your valuable comments and suggestions!

[Figure]

Figure 3 Global risk of flash drought occurrence. Risk is calculated from Eq. (4) as the probability of soil moisture dropping from 40 to 20 percentiles within 20 days or less. Similar patterns can be found by using different thresholds.

**References**

Betts, A. K., J. H. Ball, A. C. M. Beljaars, M. J. Miller, and P. A. Viterbo, 1996: The land surface-atmosphere interaction: A review based on observational and global modeling perspectives. *J. Geophys. Res.-Atmospheres*, **101**, 7209–7225, https://doi.org/10.1029/95jd02135.

Cerasoli, S., J. Yin, and A. Porporato, 2021: Cloud cooling effects of afforestation and reforestation at midlatitudes. *Proc. Natl. Acad. Sci.*, **118**, https://doi.org/10.1073/pnas.2026241118.

Christian, J. I., J. B. Basara, E. D. Hunt, J. A. Otkin, J. C. Furtado, V. Mishra, X. Xiao, and R. M. Randall, 2021: Global distribution, trends, and drivers of flash drought occurrence. *Nat. Commun.*, **12**, 6330, https://doi.org/10.1038/s41467-021-26692-z.

Daly, E., and A. Porporato, 2006: Impact of hydroclimatic fluctuations on the soil water balance. *Water Resour. Res.*, **42**, W06401, https://doi.org/10.1029/2005WR004606.

Dirmeyer, P. A., and J. Shukla, 1994: Albedo as a modulator of climate response to tropical deforestation. *J. Geophys. Res.*, **99**, 20863, https://doi.org/10.1029/94JD01311.

Duan, J., H. Gao, and B. SCHMALFUß, 2002: Stochastic dynamics of a coupled atmosphere–ocean model. *Stoch. Dyn.*, **02**, 357–380, https://doi.org/10.1142/S0219493702000467.

Eckmann, J.-P., and D. Ruelle, 1985: Ergodic theory of chaos and strange attractors. *The theory of chaotic attractors*, Springer, 273–312.

Findell, K. L., and E. A. B. Eltahir, 2003: Atmospheric Controls on Soil Moisture–Boundary Layer Interactions. Part I: Framework Development. *J. Hydrometeorol.*, **4**, 552–569, https://doi.org/10.1175/1525-7541(2003)004<0552:ACOSML>2.0.CO;2.

Kleidon, A., and M. Heimann, 1999: Deep-rooted vegetation, Amazonian deforestation, and climate: results from a modelling study. *Glob. Ecol. Biogeogr.*, **8**, 397–405, https://doi.org/10.1046/j.1365-2699.1999.00150.x.

Laio, F., A. Porporato, L. Ridolfi, and I. Rodriguez-Iturbe, 2001: Plants in water-controlled ecosystems: active role in hydrologic processes and response to water stress: II. Probabilistic soil moisture dynamics. *Adv. Water Resour.*, **24**, 707–723, https://doi.org/10.1016/S0309-1708(01)00005-7.

Last, G., and M. Penrose, 2017: *Lectures on the Poisson process*. Cambridge University Press,.

Li, J., Z. Wang, X. Wu, J. Chen, S. Guo, and Z. Zhang, 2020: A new framework for tracking flash drought events in space and time. *CATENA*, **194**, 104763, https://doi.org/10.1016/j.catena.2020.104763.

Nijzink, R., and Coauthors, 2016: The evolution of root-zone moisture capacities after deforestation: a step towards hydrological predictions under change? *Hydrol. Earth Syst. Sci.*, **20**, 4775–4799, https://doi.org/10.5194/hess-20-4775-2016.

O'Connor, J., M. J. Santos, K. T. Rebel, and S. C. Dekker, 2019: The influence of water table depth on evapotranspiration in the Amazon arc of deforestation. *Hydrol. Earth Syst. Sci.*, **23**, 3917–3931, https://doi.org/10.5194/hess-23-3917-2019.

Rodríguez-Iturbe, I., and A. Porporato, 2004: *Ecohydrology of water-controlled ecosystems: soil moisture and plant dynamics*. Cambridge University Press,.

Runyan, C. W., P. D'Odorico, and D. Lawrence, 2012: Physical and biological feedbacks of deforestation. *Rev. Geophys.*, **50**, https://doi.org/10.1029/2012RG000394.

Salazar, A., J. Katzfey, M. Thatcher, J. Syktus, K. Wong, and C. McAlpine, 2016: Deforestation changes land–atmosphere interactions across South American biomes. *Glob. Planet. Change*, **139**, 97–108, https://doi.org/10.1016/j.gloplacha.2016.01.004.

Shukla, J., C. Nobre, and P. Sellers, 1990: Amazon Deforestation and Climate Change. *Science*, **247**, 1322–1325, https://doi.org/10.1126/science.247.4948.1322.

Stefanon, M., F. D'Andrea, and P. Drobinski, 2012: Heatwave classification over Europe and the Mediterranean region. *Environ. Res. Lett.*, **7**, 014023, https://doi.org/10.1088/1748-9326/7/1/014023.

Stott, P. A., D. A. Stone, and M. R. Allen, 2004: Human contribution to the European heatwave of 2003. *Nature*, **432**, 610–614, https://doi.org/10.1038/nature03089.

Tuttle, S., and G. Salvucci, 2016: Empirical evidence of contrasting soil moisture–precipitation feedbacks across the United States. *Science*, **352**, 825–828, https://doi.org/10.1126/science.aaa7185.

Veldkamp, E., M. Schmidt, J. S. Powers, and M. D. Corre, 2020: Deforestation and reforestation impacts on soils in the tropics. *Nat. Rev. Earth Environ.*, **1**, 590–605, https://doi.org/10.1038/s43017-020-0091-5.

Yin, J., J. D. Albertson, J. R. Rigby, and A. Porporato, 2015: Land and atmospheric controls on initiation and intensity of moist convection: CAPE dynamics and LCL crossings. *Water Resour. Res.*, **51**, 8476–8493, https://doi.org/10.1002/2015wr017286.

---

## Author Response (AR1)

**DEPARTMENT OF HYDROMETEOROLOGY**
NANJING UNIVERSITY OF INFORMATION SCIENCE & TECHNOLOGY,
NANJING, JIANGSU, CHINA, 210044

JUN YIN
PROFESSOR
DEPARTMENT OF HYDROMETEOROLOGY

TEL +86-25-58731556
EMAIL: JUN.YIN@NUIST.EDU.CN

Nanjing, February 1, 2023

Dr. Peleg
University of Lausanne
Institute of Earth Surface Dynamics
Switzerland

Dear Professor Peleg,

We thank you and the reviewers for the thorough evaluation of our paper and the constructive comments. Following these suggestions, we have thoroughly revised the previous manuscript (hess-2022-313) as detailed in the next pages. In particular, we made the following main changes, motivated by the reviewers' comments:

- We calculated the variance and cumulative density function of first passage time to account for the uncertainties and risks of flash drought occurrence.
- We provided a new discussion section to address the impacts of deforestation and heatwaves on flash drought.

We hope that our manuscript is now more suitable for publication in *Hydrology and Earth System Sciences*. We thank you again and remain at your disposal for further questions and comments.

Sincerely,

**Response to Reviewers**

**Reviewer comment (italicized) is followed by a response.**

*Reviewers' comments:*

*Reviewer #1 (Remarks to the Author):*

*General comments:*

*The manuscript is written in good English and overall is well structured. The authors provide*

*extensive literature review and good methodological description*

  We thank the reviewer for the positive comments and encouragement.

• *Please improve your code documentation and comment.*

  The code documentation has been revised. A screenshot of the documentation is reported below and further information can be accessed from "github.com/yxshot/MFPT".

[Figure]

**Figure 1** The screenshot of the documentation.

*It is my opinion that the manuscript has no major technical flaws. Nevertheless, our recommendation is for Minor Reviews.*

*Specific comments:*

*Fig 4. Please improve the colour scheme, as the points in New York and Heyuan are barely visible.*

Thank you for pointing this out. We used a new color scheme and also provided two insets for New York and Heyuan. The new map was updated in the revised manuscript and is reported below.

[Figure]

**Figure 2** Global distribution of mean first passage time (MFPT) in summer. The two points marked in red are New York State, USA and Heyuan City, China. The gray areas are hyper-arid regions, other colored areas are those where MFPT of soil moisture dropping from 40 to 20 percentiles in less than 100 days. Desert regions (grey areas) are excluded in this analysis.

*Fig 4. Why did you use a limit of 100 days in the scale? Normally flash droughts intensification period is limited to up to 30 days (Osman et al, 2021; Ford and Laosier, 2015, Lisonbee et al, 2021).*

Good point. We cited these references to clarify that flash drought intensification is often within a month. However, it is still possible to have flash drought for large MFPT, which only tells the long-term averages of the intensification period. To address this point, we also derived the variance of the first passage time (VFPT) as

$$\sigma^2_{x_1 \downarrow x_2} = \int_{t_{min}}^{\infty} (t_{x_1 \downarrow x_2} - \bar{t}_{x_1 \downarrow x_2})^2 f(t_{x_1 \downarrow x_2}) dt_{x_1 \downarrow x_2}$$

$$= (t_{min} - \bar{t}_{x_1 \downarrow x_2})^2 e^{-\lambda t_{min}} + (1 - e^{-\lambda t_{min}}) \left[ 2\beta^{-2} + (t_{min} - \bar{t}_{x_1 \downarrow x_2})(t_{min} + 2\beta^{-1} - \bar{t}_{x_1 \downarrow x_2}) \right].$$

As shown in Figure 3 below, VFPT has similar spatial patterns as MFPT. The limit is set to 100 days in order to show as much as possible the areas where flash droughts are likely to occur. We have added this VFPT map in the supplementary material.

[Figure]

**Figure 3** Global distribution of variance of the mean first passage time (VFPT) (units: day2).

*Fig 4. By using the metric of Mean First Passage Time (MFPT), some areas end up showing no actual flash droughts. Please consider showing the 10th percentile of first passage time, that would show the expected occurrence in more areas.*

Thank you for the suggestion.

As also suggested by reviewer 2, we could evaluate the risk of flash drought in any given area. Therefore, regions with MFPT larger than 20 or 30 days still possible to have flash drought, albeit at a lower probability. The risk of flash drought can be quantified as the probability of first passage time lower than 20 or 30 days (or other thresholds), which is exactly the cumulative probability function (CDF). In our stochastic framework, CDF can be obtained by integrating Eq. (5), i.e.,

$$F(t_{x_1 \downarrow x_2}) = \int_{t_{\min}}^{t_{x_1 \downarrow x_2}} f(\tau)d\tau = \begin{cases} 0 & t_{x_1 \downarrow x_2} < t_{\min} \\ 1 - e^{-\beta(t_{x_1 \downarrow x_2} - t_{\min})} + e^{-\beta(t_{x_1 \downarrow x_2} - t_{\min}) - \lambda t_{\min}} & t_{x_1 \downarrow x_2} \geq t_{\min} \end{cases} \tag{1}$$

Using this equation, we can calculate the global risk of flash drought as shown in Figure 4, which has similar patterns as the global MFPT. Note that risk with different thresholds (e.g., 25 or 30 days) can still be obtained from Eq. (1). By inversing Eq. (1), we can also obtain the thresholds for any given risk (i.e., CDF = 0.1 for 10th percentile of first passage time as recommended by the reviewer). In the revised manuscript, we added this global risk figure to discuss the global patterns of flash drought.

[Figure]

**Figure 4** Global risk of flash drought occurrence. Risk is calculated as the probability of soil moisture dropping from 40 to 20 percentiles within 20 days or less.

*Fig 4. Please justify the very low MFPT in semi-arid regions, such as southern India, northern Namibia/Botswana and northeast Brazil.*

We are not quite sure if we understand this comment correctly.

As shown in Figure 2 above, the MFPT in southern India, northern Namibia/Botswana and northeast Brazil are quite high (i.e., close to or higher than 100 days). Aside from climate conditions, the long crossing time in these regions may be associated with deep rooting depths (see root-zone storage, $w_0$, in Figure 5 below), which act as a buffering zone to reduce the variation of soil moisture and thus increase the time for drought intensification (e.g., Laio et al. 2001). We clarified the role of rooting depths in the revised manuscript.

Please feel free to correct us if we misunderstood your comment. Thank you again!

[Figure]

**Figure 5** Global distribution of soil water storage capacity (units: mm), which used to calculate global MFPT.

*Reviewer #2 (Remarks to the Author):*
*General comment*

*This study used a stochastic water balance framework to examine the nonlinear relationship between the timing of drought and various hydrometeorological factors and identify possible flash drought events caused by lack of rainfall, high evapotranspiration, low soil water storage capacity, or a combination thereof. Indeed, there are a variety of definitions for flash drought, which has been merged as a critical sub-seasonal phenomenon with great impacts on agriculture, the economy, and society. Providing new metrics for flash drought from a stochastic perspective is certainly of great importance to our understanding of the rapid intensification of drought events. The stochastic theory is sound and straightforward, and the authors also found that flash drought also exists in humid regions such as southern China and the northeastern United States, calling for particular attention to flash drought monitoring and mitigation. And the manuscript is wellwritten and well structured, with potential publication in HESS. Below I list some points and the authors are wished to address before published.*

We are grateful to the reviewers for their positive comments and encouragement. We have used these suggestions to improve the manuscript, as described below.

*Major concerns*
*As illustrated in the text, the proposed framework measures the effect of deforestation on flash drought, but the description on this content is unclear. Soil water storage capacity does have a strong link with vegetation distribution, for example, drylands, with low NDVI, correspondingly show weak soil water storage capacity. In addition, deforestation can change hydrological and energy cycle processes, such as altering surface albedo and soil infiltration rate, which have an impact on flash drought. What is the relationship between deforestation and soil water storage capacity? Please add some specific statements. Further explaining is also needed, from my viewpoint, on how the framework measures the effect of deforestation on flash drought.*

Thank you for pointing this out. We did not explain this point very well in the original manuscript, but now we have clarified the linkage between deforestation and flash drought.

As commented by the reviewer, deforestation converts forest into cropland or savanna, possibly reducing the rooting depth and soil water storage capacity (Kleidon and Heimann 1999; O'Connor et al. 2019; Nijzink et al. 2016). As shown Figure 6 a and b (Fig. 3 in the manuscript), lower soil water storage capacity ($w_0$) tends to reduce the mean first passage time of soil moisture dropping from 40 to 20 percentiles, demonstrating the possible impacts of deforestation on flash drought.

Moreover, deforestation also tends to increases surface albedo and thus influence the surface energy balance and potential evaporation rate (Dirmeyer and Shukla 1994; Cerasoli et al. 2021), which have been considered in the stochastic framework. Smaller $E_{max}$ increases the mean first passage time and therefore reduce the likelihood of flash drought (see Figure 6 b and c).

The changes of soil properties after deforestation have been reviewed by Runyan et al. (2012) and Veldkamp et al. (2020) and many others. Such changes in soil organic content, retention curve, and infiltration rate inevitably influence the hydrological cycle and soil moisture dynamics (Laio et al. 2001). It is possible to include all these factors in the full stochastic framework (e.g., Rodríguez-Iturbe and Porporato 2004) to diagnose the impacts of deforestation on the soil properties and the rapid decline rates of soil moisture.

At even large scale, deforestation may also change surface temperature and precipitation though land-atmosphere interaction (Shukla et al. 1990; Salazar et al. 2016). Deforestation may change the partitioning of surface heat flux and influence the atmospheric boundary layer dynamics, controlling the transition from shallow to deep convection (Betts et al. 1996; Findell and Eltahir 2003; Yin et al. 2015; Tuttle and Salvucci 2016; Cerasoli et al. 2021). Lower precipitation rate corresponds to faster drop of soil moisture and higher probability of flash drought as shown in Figure 6 a and c.

We have included a new discussion section to address all these linkages between deforestation and flash drought in the revised manuscript.

[Figure]

**Figure 6** The influence of hydrometeorological factors on mean first passage time (days) of soil moisture dropping from 40 to 20 percentiles.

*Existing model simulations or satellite observations can provide daily-scale soil moisture as well, although these data are not free from biases. In comparison to traditional droughts, flash droughts are characterized by rapid development, while the rapid development of flash droughts usually occurs within days or weeks, so pentadscale hydrometeorological variables are commonly used and few studies analyzed flash droughts based on daily-scale data. The necessity to study the timing of flash drought based on the minimalist hydrological model should be further explained and discussed.*

Good point. Actually, we already did this, but we did not explain this approach very well in the previous version of the manuscript.

As commented by the reviewer, flash drought is often characterized by the pentad (5-day) average soil moisture, which may have smoother temporal evolution than the daily soil moisture. While soil moisture is modeled at daily timescale in our stochastic framework (see gray and black lines in Figure 7 top panel), the corresponding time for soil moisture dropping from 40 to 20 percentiles (first passage time, see the distribution in Figure 7 bottom panel) is NOT directly used to characterize the flash drought. Instead, the ensemble averages of the first passage time (i.e., averaged over many realizations of the stochastic processes) is much smoother than the first passage time for the given hydrometeorological condition and is used to characterize the rapid intensification of drought.

In fact, the soil moisture averaged over a long period is equivalent to ensemble average under the ergodic hypothesis, which is usually valid in a chaotic system such as the soil water dynamics driven by stochastic forcing (Eckmann and Ruelle 1985; Duan et al. 2002) at the scales considered here. In its strictest form, the ergodic hypothesis states that ensemble statistics at any given time or position are identical to the temporal or spatial statistics (mean and higher-order moments). Therefore, the crossing time of the pentad average soil moisture should asymptotically approach to the MFPT used in this study, which could provide accurate description of soil moisture dry-down process.

In the revised manuscript, we have clarified the differences between the first passage time and the mean first passage time and explicitly state that the latter is used to characterize the flash drought.

[Figure]

**Figure 7** (top) numerical simulation of water balance for relative soil moisture x dropping from 40 to 20 percentiles, and (bottom) the corresponding distribution of first passage time (sample size of 1000).

*One more point I concern is that the framework can measure the effect of evapotranspiration (E) on flash drought, yet there is difference between potential evapotranspiration (PET) and E, for example*

*for moisture-limited dry lands. I don't know did the authors measure the difference between E and PET on the results in Figure 3? In addition, the change in E is related to heatwave, while other factors (such as change in leaf area index and solar radiation) can also impact E. I suggest adding some discussion, in particular, on the difference between E and PET.*

Thank you for the valuable suggestions.

Yes, the differences between $E$ and PET were considered in Figure 3. While these differences have been briefly discussed in the original manuscript (Line 58), it is not very clear and has been explicitly addressed in the revised manuscript.

In the water balance model, $E$ is assumed to be a function of soil moisture and potential evapotranspiration, i.e.,

$$E = f(E_{max}, x) = xE_{max} ,$$ (2)

where the last equality assumes $E$ linearly increase from 0 for $x = 0$ to $E_{max}$ for $x = 1$ in the minimalist framework. It should be noted that more general form of $f(E_{max}, x)$ can still be solved analytically for the mean first passage time. Therefore, we can model evapotranspiration with different soil water thresholds such as the wilting point, onset of the soil water stress, and field capacity in the more general stochastic framework to explore the mean first passage time and the flash drought.

Moreover, as commented by the reviewer, heatwave, leaf area index, and solar radiation also influence flash drought, which have been thoroughly discussed in the revised manuscript. Specifically, we used Penman equation to introduce the potential evapotranspiration as

$$E_{max} = \underbrace{\frac{\Delta}{\rho_w \lambda_w (\Delta + \gamma)} Q}_{E_e} + \underbrace{\frac{\gamma}{\rho_w (\Delta + \gamma)} \left( \frac{\varepsilon}{p_0} \rho g_a \text{VPD} \right)}_{E_v} ,$$ (3)

where $E_e$ is equilibrium evapotranspiration, $E_v$ is the evapotranspiration due to drying power of the air, $\Delta$ is the slope of the saturation vapor pressure curve (a nonlinear function of air temperature), $\gamma$ is psychrometric constant, $\lambda_w$ is latent heat of water vaporization, $Q$ is available surface energy, $\varepsilon$ is the ratio of the gas constant for dry air to that of water vapor, $p_0$ is near-surface air pressure, $\rho$ is air density, $\rho_w$ water density, $g_a$ is aerodynamic conductance, and VPD is vapor pressure deficit. Heatwave is often accompanied with high temperature and strong solar radiation (Stott et al. 2004), which tend to increase $E_e$; dry or moist heatwaves may also have abnormal VPD (Stefanon et al. 2012), which may influence $E_v$. Vegetation with larger leaf area index tends to have higher surface roughness, resulting in larger $g_a$ and $E_v$. Therefore, heatwave, leaf area index, and solar radiation influence the potential evapotranspiration, which further controls the soil moisture dynamics and the drought occurrence.

We have added the Penman equation in the supplementary material and incorporated these discussions in the revised manuscript.

*Minor concerns*

*Aside from soil moisture, evaporation deficit (PET-ET) or evaporative stress ratio (ET/PET) is often closely monitored to quantify the intensification of flash drought. It would be useful also to provide a more general framework to consider these variables (or at least these variables should be acknowledged).*

These are valid points; thank you!

As commented by the reviewer, evaporative stress ratios ($E/E_{max}$) or evaporation deficit ($E-E_{max}$) were also used to characterize flash droughts (e.g., Li et al. 2020; Christian et al. 2021). In the minimalist framework with $E = xE_{max}$, evaporative stress ratio is already equivalent to $x$, which has been discussed in the original manuscript. In the more general form, when modeling evaporation as a function soil potential evaporation and soil moisture, we can further rewrite these two metrics as

$$\begin{cases} \dfrac{E}{E_{max}} = \dfrac{f(E_{max}, x)}{E_{max}} \\ E - E_{max} = f(E_{max}, x) - E_{max} \end{cases}, \tag{4}$$

which are functions of $E_{max}$ and $x$. If we assume the daily variations of $E_{max}$ have limited impacts on soil water balance (Daly and Porporato 2006), we can treat evaporative stress ratios or deficit as the derived distributions of soil moisture, allowing us to link the corresponding percentiles and crossing properties to these for soil moisture.

In the revised the manuscript, we have discussed these metrics in the stochastic framework for diagnosing of flash drought.

*Line 75: the example given in Fig. 2c clearly shows an exponential tail. Can we still have exponential distribution for parameters with different values? This should be explored.*

We still have exponential tails, which were explored by linearly fitting the logarithmic of the tails for different parameters. From our numerical simulations (n =1000) in the parameter space of Figure 2, the correlation coefficients between $log(p(t_{x_1 \downarrow x_2}))$ and $t_{x_1 \downarrow x_2}$ for $t > t_{min}$ are close to -1 with mean value of -0.94 and standard deviation of 0.03, suggesting these tails could be properly described as exponential. These results have been reported in the revised manuscript.

*Abstract should also emphasize the probabilistic structure of the first passage time, which is the benefit of the stochastic framework.*

Thank you for your suggestion. Particularly, we stressed that the stochastic water balance framework can be used to describe the probability of the timing for soil moisture dropping from a higher level to a lower one.

*Line 13: period is missed after the citation.*

Corrected.

*I think there should be minus sign in front of Eq. (4).*

Thank you for your reminder.

*Line 74: The atom probability of no rainfall is not trivial. Please provide references or details of its derivation.*

The reference of Last and Penrose (2017) has been provided in the revised manuscript.

*How to calculate the rainfall frequency and average depth. Please clarify.*

The time series of the daily precipitation in the boreal summer of 2009-2018 was obtained from the Global Precipitation Climatology Project (GPCP). From these records, we calculated the rainfall frequency as the proportion of raining days and rainfall depth as the average of daily rainfall depth (excluding days without rainfall). This has been clarified in the beginning of the Sec. 3.2.

*The information provided by each picture is seemingly not enough. Can you add more information, please?*

Thank you for the reminder.
In the revised manuscript, we have provided more information in the captions to specify all parameters and data sources used in the figure.

*Discussion chapters should be added to enrich the content*

Thank you for pushing us to improve the presentation of the manuscript. In the revised reversion, we have added a new discussion section to address the potential impacts from heatwave, deforestation, solar radiation, and land-atmosphere interaction on the flash drought. We also included the Penman equation in the supplementary material to explain that larger potential evapotranspiration reduces the time for soil moisture dropping from higher level to a lower one.

*I think the explanation of "timing of drought" in the text is slightly vague, which may further affect the readers' understanding of the drought risk mentioned in the study. Please add some explanations for this concept.*

To be consist with previous studies (e.g., Li et al. 2020; Christian et al. 2021), we used 'rapid decline rate of soil moisture' to characterize the flash drought throughout the text.

*Could you point out the numerical interval of timing of drought with high risk of flash drought?*

This is an interesting question. Thank you!
As suggested in previous studies (e.g., Otkin et al., 2016; Ford and Labosier, 2017; Basara et al., 2019; Nguyen et al., 2019; Zhang et al., 2022), the flash drought is often characterized as the soil moisture dropping from 40 to 20 percentiles within 20 days. Therefore, the risk of flash drought can be quantified as the probability of first passage time lower than 20 days (or other thresholds), which is exactly the cumulative probability function (CDF). In our stochastic framework, CDF can be obtained by integrating Eq. (5), i.e.,

$$F(t_{x_1 \downarrow x_2}) = \int_{t_{\min}}^{t_{x_1 \downarrow x_2}} f(\tau)d\tau = \begin{cases} 0 & t_{x_1 \downarrow x_2} < t_{\min} \\ 1 - e^{-\beta(t_{x_1 \downarrow x_2} - t_{\min})} + e^{-\beta(t_{x_1 \downarrow x_2} - t_{\min}) - \lambda t_{\min}} & t_{x_1 \downarrow x_2} \geq t_{\min} \end{cases} \quad (5)$$

Using this equation, we can calculate the global risk of flash drought as shown in Figure 8, which has similar patterns as the global MFPT (i.e., Figure 4 in the original manuscript). Note that risk with different thresholds (e.g., 25 or 30 days) can still be obtained from Eq. (4).

By inversing Eq. (4), we can also obtain the thresholds for any given risk (i.e., probability). In the revised manuscript, we added this global risk figure to explain the unique feature of our methods.

We thank you again for your valuable comments and suggestions!

[Figure]

**Figure 8** Global risk of flash drought occurrence. Risk is calculated from Eq. (4) as the probability of soil moisture dropping from 40 to 20 percentiles within 20 days or less. Similar patterns can be found by using different thresholds.

**References**

Betts, A. K., J. H. Ball, A. C. M. Beljaars, M. J. Miller, and P. A. Viterbo, 1996: The land surface-atmosphere interaction: A review based on observational and global modeling perspectives. *J. Geophys. Res.-Atmospheres*, **101**, 7209–7225, https://doi.org/10.1029/95jd02135.

Cerasoli, S., J. Yin, and A. Porporato, 2021: Cloud cooling effects of afforestation and reforestation at midlatitudes. *Proc. Natl. Acad. Sci.*, **118**, https://doi.org/10.1073/pnas.2026241118.

Christian, J. I., J. B. Basara, E. D. Hunt, J. A. Otkin, J. C. Furtado, V. Mishra, X. Xiao, and R. M. Randall, 2021: Global distribution, trends, and drivers of flash drought occurrence. *Nat. Commun.*, **12**, 6330, https://doi.org/10.1038/s41467-021-26692-z.

Daly, E., and A. Porporato, 2006: Impact of hydroclimatic fluctuations on the soil water balance. *Water Resour. Res.*, **42**, W06401, https://doi.org/10.1029/2005WR004606.

Dirmeyer, P. A., and J. Shukla, 1994: Albedo as a modulator of climate response to tropical deforestation. *J. Geophys. Res.*, **99**, 20863, https://doi.org/10.1029/94JD01311.

Duan, J., H. Gao, and B. SCHMALFUß, 2002: Stochastic dynamics of a coupled atmosphere–ocean model. *Stoch. Dyn.*, **02**, 357–380, https://doi.org/10.1142/S0219493702000467.

Eckmann, J.-P., and D. Ruelle, 1985: Ergodic theory of chaos and strange attractors. *The theory of chaotic attractors*, Springer, 273–312.

Findell, K. L., and E. A. B. Eltahir, 2003: Atmospheric Controls on Soil Moisture–Boundary Layer Interactions. Part I: Framework Development. *J. Hydrometeorol.*, **4**, 552–569, https://doi.org/10.1175/1525-7541(2003)004<0552:ACOSML>2.0.CO;2.

Kleidon, A., and M. Heimann, 1999: Deep-rooted vegetation, Amazonian deforestation, and climate: results from a modelling study. *Glob. Ecol. Biogeogr.*, **8**, 397–405, https://doi.org/10.1046/j.1365-2699.1999.00150.x.

Laio, F., A. Porporato, L. Ridolfi, and I. Rodriguez-Iturbe, 2001: Plants in water-controlled ecosystems: active role in hydrologic processes and response to water stress: II. Probabilistic soil moisture dynamics. *Adv. Water Resour.*, **24**, 707–723, https://doi.org/10.1016/S0309-1708(01)00005-7.

Laio, F., A. Porporato, C. P. Fernandez-Illescas, and I. Rodriguez-Iturbe, 2001: Plants in water-controlled ecosystems: active role in hydrologic processes and response to water stress IV. Discussion of real cases. *Adv. Water Resour.*, 18.

Last, G., and M. Penrose, 2017: *Lectures on the Poisson process*. Cambridge University Press,.

Li, J., Z. Wang, X. Wu, J. Chen, S. Guo, and Z. Zhang, 2020: A new framework for tracking flash drought events in space and time. *CATENA*, **194**, 104763, https://doi.org/10.1016/j.catena.2020.104763.

Nijzink, R., and Coauthors, 2016: The evolution of root-zone moisture capacities after deforestation: a step towards hydrological predictions under change? *Hydrol. Earth Syst. Sci.*, **20**, 4775–4799, https://doi.org/10.5194/hess-20-4775-2016.

O'Connor, J., M. J. Santos, K. T. Rebel, and S. C. Dekker, 2019: The influence of water table depth on evapotranspiration in the Amazon arc of deforestation. *Hydrol. Earth Syst. Sci.*, **23**, 3917–3931, https://doi.org/10.5194/hess-23-3917-2019.

Rodríguez-Iturbe, I., and A. Porporato, 2004: *Ecohydrology of water-controlled ecosystems: soil moisture and plant dynamics*. Cambridge University Press,.

Runyan, C. W., P. D'Odorico, and D. Lawrence, 2012: Physical and biological feedbacks of deforestation. *Rev. Geophys.*, **50**, https://doi.org/10.1029/2012RG000394.

Salazar, A., J. Katzfey, M. Thatcher, J. Syktus, K. Wong, and C. McAlpine, 2016: Deforestation changes land–atmosphere interactions across South American biomes. *Glob. Planet. Change*, **139**, 97–108, https://doi.org/10.1016/j.gloplacha.2016.01.004.

Shukla, J., C. Nobre, and P. Sellers, 1990: Amazon Deforestation and Climate Change. *Science*, **247**, 1322–1325, https://doi.org/10.1126/science.247.4948.1322.

Stefanon, M., F. D'Andrea, and P. Drobinski, 2012: Heatwave classification over Europe and the Mediterranean region. *Environ. Res. Lett.*, **7**, 014023, https://doi.org/10.1088/1748-9326/7/1/014023.

Stott, P. A., D. A. Stone, and M. R. Allen, 2004: Human contribution to the European heatwave of 2003. *Nature*, **432**, 610–614, https://doi.org/10.1038/nature03089.

Tuttle, S., and G. Salvucci, 2016: Empirical evidence of contrasting soil moisture–precipitation feedbacks across the United States. *Science*, **352**, 825–828, https://doi.org/10.1126/science.aaa7185.

Veldkamp, E., M. Schmidt, J. S. Powers, and M. D. Corre, 2020: Deforestation and reforestation impacts on soils in the tropics. *Nat. Rev. Earth Environ.*, **1**, 590–605, https://doi.org/10.1038/s43017-020-0091-5.

Yin, J., J. D. Albertson, J. R. Rigby, and A. Porporato, 2015: Land and atmospheric controls on initiation and intensity of moist convection: CAPE dynamics and LCL crossings. *Water Resour. Res.*, **51**, 8476–8493, https://doi.org/10.1002/2015wr017286.